# Plasma Lipid Profile Among Perimenopausal Latvian Women in Relation to Dietary Habits

**DOI:** 10.3390/nu17020243

**Published:** 2025-01-10

**Authors:** Svetlana Aleksejeva, Laila Meija, Maksims Zolovs, Inga Ciprovica

**Affiliations:** 1Faculty of Agriculture and Food Technology, Latvia University of Life Sciences and Technologies, LV-3001 Jelgava, Latvia; inga.ciprovica@lbtu.lv; 2Department of Rehabilitation, Rīga Stradiņš University, LV-1007 Riga, Latvia; laila.meija@rsu.lv; 3Institute of Life Sciences and Technology, Daugavpils University, LV-5401 Daugavpils, Latvia; maksims.zolovs@du.lv; 4Statistics Unit, Rīga Stradiņš University, LV-1007 Riga, Latvia

**Keywords:** cholesterol, dyslipidemia, perimenopause, cardiovascular disease, dietary habits

## Abstract

**Background/Objectives:** Hormonal changes throughout a woman’s life cycle significantly affect serum lipid levels. Alterations in the serum lipid profile can increase the risk of cardiovascular diseases (CVDs). Additionally, nutrition and dietary habits are crucial for managing dyslipidemia. The current study evaluated the association between dietary habits and plasma lipid profiles among perimenopausal women in Latvia. **Methods:** The randomized clinical trial involved perimenopausal women (*n* = 61) aged 49 ± 3 years with moderately high low-density lipoprotein cholesterol (LDL-C) levels of 3.61 ± 0.30 mmol L^−1^. A series of questionnaires were completed, including a questionnaire on the subject’s demographic, anthropometric, lifestyle, health, physical activity, and dietary factors, a 24 h food diary, a 72 h food diary, and a food-frequency questionnaire (FFQ). Blood testing was conducted for all participants, which included total cholesterol (TC), low-density lipoprotein cholesterol (LDL-C), high-density lipoprotein cholesterol (HDL-C), triglycerides (TG), alanine aminotransferase (ALAT), and glucose level analyses. **Results:** The consumption of refined sugar, honey, syrup, and jam demonstrated a strong positive association with higher levels of remnant cholesterol (β = 0.462, *p* ≤ 0.05) and non-high-density lipoprotein cholesterol (non-HDL-C) (β = 0.395, *p* ≤ 0.05). Similarly, the consumption of fruit juices is associated with increased LDL-C (β = 0.303, *p* ≤ 0.05) and non-HDL-C (β = 0.285, *p* ≤ 0.05). Conversely, higher meat and poultry consumption negatively correlates with TC levels (β = −0.290, *p* ≤ 0.05). **Conclusions:** This underscores the need for further examination to understand the impact of dietary habits on lipid profile.

## 1. Introduction

Perimenopause is the phase before menopause, marked by hormonal changes and various symptoms. It usually begins in a woman’s forties and can last four to eight years. During this time, fluctuating estrogen levels may cause irregular periods, hot flashes, night sweats, mood swings, sleep issues, and altered sexual function, reflecting the body’s adjustment to reduced ovarian function [1,2]. An initial stage usually occurs around the average age of 47, followed by a later stage that begins around the average age of 49. However, the duration of these stages and the symptoms experienced can vary significantly [3].

Perimenopause is a critical transitional phase impacting women’s health and quality of life, with challenges like depression and long-term risks to bone density, urogenital health, and cholesterol levels [4].

During this phase, alterations in the plasma lipid profile and estrogen deficiency increase the risk of CVDs by affecting cholesterol metabolism and promoting dyslipidemia, endothelial dysfunction, and vascular inflammation [5,6,7,8].

CVDs such as cerebrovascular disease, rheumatic heart disease, and coronary heart disease are the primary cause of mortality worldwide. As reported by the World Health Organization (WHO), CVDs are responsible for 17.9 million deaths each year [9]. In 2019, the United States statistics revealed that CVDs were the cause of 48.1% of all deaths related to CVDs among women [10]. In Europe, CVDs account for 49% of all deaths among women [11]. In 2021, the primary causes of death in Latvia were circulatory diseases, especially ischemic heart disease and stroke—in total, 49.4% [12].

Reducing reproductive hormone levels significantly affects CVD risk factors. It causes harmful changes in the lipid profile, such as increased levels of TC, LDL-C, and triglycerides (TG), while simultaneously lowering HDL-C levels [13,14].

Dietary patterns are key factors in CVD risk, making their examination essential for prevention and risk reduction [15]. Lifestyle changes and a healthy diet rich in fruits, vegetables, whole grains, and lean proteins while limiting added sugars, saturated fats, and sodium can reduce CVD risk [16].

Past studies have investigated this link across various populations, concentrating on the relationship between dietary habits and mortality from chronic diseases [17,18].

It has been consistently found that a high-quality diet significantly lowers the risk of both all-cause and cause-specific mortality. Regarding specific dietary patterns, those considered healthy, characterized notably by high consumption of fruits and vegetables, whole grains, and legumes, have been linked to a decreased mortality risk from chronic diseases [19,20,21].

In Latvia, statistical data demonstrate that women, on average, transition from an average or underweight status to overweight or obese around the age of 45, reflecting population trends rather than longitudinal individual changes. Among women aged 45 to 64, 66.1% are either overweight or obese [22]. There are no statistical data on the number of women deaths from CVDs, but the prevalence of excess body weight significantly increases mortality and morbidity risk. Approximately 70% of deaths related to high BMI are caused by CVDs, with over 60% of those deaths occurring among obese individuals [23].

This study investigated the association between dietary habits and the plasma lipid profile among perimenopausal women in Latvia. It contributes to the broader scientific literature on women’s health, nutrition, and CVD risk management during the perimenopausal period. Despite the scientific understanding of the importance of nutrition in managing the plasma lipid profile, more data are needed, specifically on the Latvian population.

## 2. Materials and Methods

### 2.1. Design, Time, and Location of the Research and Sample Size

The randomized clinical trial was conducted in Latvia from October 2023 to June 2024. Participants were recruited based on convenience, utilizing gynecological practices, general practitioners, hospitals, and social media.

### 2.2. Inclusion and Exclusion Criteria for the Participants

The inclusion criteria for the participants were:Reside in Latvia.Women.Perimenopause symptoms (with vasomotor symptoms (hot flushes and night sweats) and infrequent periods).45–55 years.Moderately elevated LDL-C (3.01–4.12 mmol L^−1^) within the last 6 months.Stable body weight (+/−2 kg) in the previous 6 months.

The exclusion criteria were:Noncompliance with the inclusion criteria.History of primary hyperlipidemia.CVDs (ischemic heart disease, myocardial infarction, stroke, history of chronic heart failure, angina pectoris, or arrhythmia).Endocrine diseases (type 1 or type 2 diabetes mellitus, thyroid pathology (hypothyroidism or hyperthyroidism), or adrenal disease (Cushing’s syndrome or secondary renal failure)).Liver diseases (non-alcoholic fatty liver disease (NATAS), alcoholic fatty liver disease (ATAS), hepatitis, cirrhosis, or chronic liver failure), chronic kidney disease, or a history of gastrointestinal surgery.Hormonal therapy.Immunosuppressive therapy.Pharmacological treatment of hyperlipidemia or hypertension therapy (drugs or food supplements).

### 2.3. Ethical Considerations

This study was conducted in compliance with the Declaration of Helsinki and received approval from the Ethics Committee of Rīga Stradiņš University (protocol code 2-PĒK-4/513/2022, approved on 23 November 2022). Signed informed consent was obtained from all participants involved in the study.

### 2.4. Description of the Study Design

The study began with recruiting participants based on specific inclusion and exclusion criteria. Out of the 1,096 women who initially showed interest in the study, only 37.7% (*n* = 404) met the requirements. Eligible participants received a detailed study invitation, with 51.7% (*n* = 209) expressing willingness to proceed. Consent forms were signed and returned by 54.1% (*n* = 113) of these women.

Following consent, participants completed a questionnaire on demographic, anthropometric, lifestyle, health, physical activity, and dietary factors, with 95.6% (*n* = 108) remaining eligible. Anthropometric data were self-reported and used to calculate the Body Mass Index (BMI) and the waist-to-hip ratio. The structure of the Food Frequency Questionnaire was adapted using a previously developed questionnaire of the Latvian National Health Behaviour Survey on the general population (Appendix A) [24].

Baseline tests were conducted, including measurements of TC, LDL-C, HDL-C, TG, ALAT, and glucose levels. Blood tests were performed in the accredited laboratory Centrālā Laboratorija Ltd. (Rīga, Latvia, accreditation certificate LATAK-M-434-04-2011) with 100 branches in the Baltics. The enzymatic color reaction method was used for lipid profile measurements in serum, the kinetic reaction method for ALAT determination in serum, and the hexokinase method for plasma glucose estimation. Remnant cholesterol and non-HDL-C were calculated by the Centrālā Laboratorija as a part of standard procedure. Remnant cholesterol is the cholesterol in triglyceride-rich lipoproteins, including very low-density lipoproteins (VLDL) and intermediate-density lipoproteins (IDL). Non-HDL-C includes all atherogenic lipoproteins like LDL-C, VLDL-C, and IDL-C. It is determined by taking the TC and subtracting HDL-C from it.

The randomization of 62 (sixty-two) participants was performed using block randomization to balance the intervention and control groups. As participants joined, they were assigned to the next group in the randomization list, completing one block before moving to the next.

Participants completed a self-administered 24 h and 72 h food diary over three consecutive days. Food diaries were completed after the researcher provided guidance via a video tutorial and question-and-answer sessions. The food diary was administered with an atlas of food products and food portions of the Food Safety and Animal Health Research Institute (BIOR) to assess dietary intake [25].

To evaluate the study participants’ dietary habits, each participant completed a Food Frequency Questionnaire (FFQ) assessing food and nutrient intake over the past year with the researcher’s assistance. The interviews were conducted online

Combining FFQ with a 24 h food diary enhances the accuracy of dietary intake estimates [26]. Moreover, 24 h and 72 h food diaries help to identify any habits that may later be omitted in self-completed records and to verify whether the frequency of marked foods or quantities corresponds to those recorded in the dietary records, allowing one to assess the attentiveness and accuracy of the participants.

The questionnaire comprised 260 food products, beverages, and dietary supplements, categorized into 26 groups (Appendix A). The average consumption for each of the food groups was then analyzed.

Participants indicated their frequency of consumption for each food item by selecting one of the following options: “never” to “6 times a day or more” for each product with an indicated portion size.

The nutritional data were processed in BIOR using Microsoft Dynamics AX 2009. This research provides information on food intake data exclusively from the FFQ.

In total, 61 participants completed the study.

Descriptive data are reported as means and standard deviations for continuous variables, while categorical variables are displayed as counts and percentages. Linear regression analysis evaluated the associations between plasma lipid profile parameters and food group intake. An adjusted association (age, BMI, and waist-to-hip ratio) with the dependent variable (plasma lipid profile parameters) was included for each food group. A *p*-value of ≤0.05 was considered statistically significant for all statistical tests. Heat-map colors are based on adjusted standardized regression coefficients β. Jamovi (2.6) statistical software (Jamovi Project, Sydney, Australia) was used for data analysis.

## 3. Results

### 3.1. Baseline Characteristics of the Study Participants

Table 1 presents the baseline characteristics of the study participants (*n* = 61) who completed the study, detailing key anthropometrics, alcohol consumption, smoking status, light physical activity status, and number of meals. The average age of the participants was 49 ± 3 years (mean ± SD). Participants had a mean BMI of 25.05 ± 4.92 kg m^−2^, which falls into the overweight category (25.00 to 29.90 kg m^−2^) according to standard BMI classifications.

A significant proportion of the study participants reported alcohol usage, with 59.02% indicating regular consumption. Smoking prevalence, however, was notably low, with only 4.92% of participants identifying as smokers.

Table 2 presents the biochemical profile of participants, revealing an average TC level of 5.93 ± 0.54 mmol L^−1^, which is above optimal thresholds. HDL-C averaged 1.85 ± 0.46 mmol L^−1^ and LDL-C averaged 3.61 ± 0.30 mmol L^−1^, which is moderately high. TG levels averaged 1.15 ± 0.64 mmol L^−1^, while glucose levels were relatively well-controlled at 4.67 ± 0.67 mmol L^−1^, remaining within a normal range for fasting glucose. As represented by ALAT, liver enzyme levels showed a mean value of 21.00 ± 10.21 U L^−1^, suggesting normal liver function in the participants.

### 3.2. Dietary Habits Among the Study Participants

Table 3 summarizes the average daily intake of various food products and beverages among the study participants based on information from the FFQ, providing insights into dietary habits.

Grains and grain-based products comprised a significant portion of the diet, with high consumption levels noted. Meat and poultry also played an important role, complemented by bread, an essential food in the diet. Potatoes, recognized as a typical staple food, were consumed consistently. However, the intake of meat offal and other processed meat products was relatively low, indicating consumption on an occasional basis.

Dairy products served as a significant source of nutrition, while plant-based dairy substitutes were consumed in lower amounts. The daily average intake of vegetables and fruits was almost equal. On the other hand, the consumption of fish and fish products was very low.

Among animal-based foods, milk and milk products were widely consumed, exceeding the intake of meat, fish, and eggs. Among plant-based foods, fruits and berries were the most popular, followed by vegetables. Legumes, however, showed relatively low intake.

Water dominated the participants’ diet, followed by coffee and tea.

The average daily energy and macronutrient intakes among participants are shown in Table 4.

Participants’ average daily energy and macronutrient intakes meet the recommended daily intake guidelines for adults in Latvia. Total fats contributed the most significantly, accounting for 42.41% of daily energy. Carbohydrates followed at 35.35%, while proteins contributed 19.58%.

SFA comprised a substantial part of total fat intake, accounting for 13.52% of total energy consumption.

The daily consumption of monounsaturated fats (MUFA) reached 16.12% of the total energy intake, and polyunsaturated fats (PUFA) reached 7.37% of the total energy intake.

### 3.3. Dietary Choices and Lipid Profile

Data from the food frequency questionnaire are compiled in Figure 1.

Figure 2 presents a heat map of standardized regression coefficients (β) obtained from linear regression analyses between average food group intakes and lipid profile parameters from the food-frequency questionnaires.

Foods from the group “Sugars and related food” demonstrated a strong association with higher levels of remnant cholesterol (β = 0.462, *p* ≤ 0.05) and non-HDL-C (β = 0.395, *p* ≤ 0.05). Similarly, the consumption of juices was associated with increased LDL-C (β = 0.303, *p* ≤ 0.05) and non-HDL-C (β = 0.285, *p* ≤ 0.05).

Sweets exhibited a comparable trend, where higher consumption correlated with elevated levels of remnant cholesterol (β = 0.256, *p* ≤ 0.05) and non-HDL-C (β = 0.247, *p* ≤ 0.05).

Conversely, higher meat and poultry consumption negatively correlated with TH levels (β = −0.290, *p* ≤ 0.05).

Foods from the group “Vegetables” demonstrated a non-statistically significant (*p* = 0.407) correlation with non-HDL-C levels.

## 4. Discussion

The mean BMI of participants indicates overweight status and is further supported by a mean waist circumference of 84.34 ± 11.26 cm, commonly associated with cardiovascular and metabolic risk. A waist-to-hip ratio of 0.81 ± 0.07 is within the normal range [29]. Although it falls within the normal range, the elevated BMI and waist circumference highlight the potential need for targeted interventions to mitigate these risks.

These results indicate that consistent engagement in physical activities can lead to positive alterations in lipid profile—it promotes an increase in HDL-C levels while reducing TG. Physical activity has advantageous effects on TC and LDL-C [30,31]. Participants were highly engaged in daily physical activities, with 83.61% reporting regular exercise, emphasizing a generally active lifestyle. Despite the high level of activity reported by the participants, their lipid profile underscores the importance of incorporating moderate to vigorous physical activities, which could yield even more significant benefits regarding lipid profile improvements.

A significant majority of participants in the study, accounting for 73.77%, reported consuming two to three main meals per day. This pattern suggests a relatively consistent meal structure among most individuals. In contrast, approximately 19.67% of respondents indicated that they eat fewer than two main meals daily in the case of smaller portion sizes. This could point to various factors, such as irregular eating habits or possible dietary restrictions that affect their meal frequency. Only a small number of respondents (6.56%) stated that they consume four to five main meals within the same time frame. However, such frequency is uncommon among the participants.

The literature data on eating frequency present mixed findings. Some studies suggest that a higher frequency of meals and snacks may be linked to a reduced risk of CVDs, although this relationship remains complex and not universally established [32,33]. A small minority (3.28%) reported completely abstaining from snacks daily, highlighting that most individuals engage in some form of snacking. The most prevalent snacking frequency was two daily snacks, favored by 37.70% of the participants. Close behind, 32.79% indicated that they typically enjoy one snack daily. Furthermore, a notable 19.67% of the surveyed participants engaged in snacking three times a day, while a smaller group, 6.56%, admitted to having four or more snacks within 24 h.

The Ministry of Health of the Republic of Latvia strongly advises that individuals aim to incorporate a minimum of 500 g of diverse vegetables, fruits, and berries into their daily diets. This recommendation emphasizes the consumption of 300 g of vegetables and 200 g of fruits or berries to promote optimal health [34]. Research results have indicated a negative correlation between vegetable intake and the likelihood of developing CVDs [35,36,37,38]. Despite this guidance, the current results reveal that the average daily vegetable consumption among the participants falls short of the suggested amount, highlighting the significance of vegetables in maintaining heart health.

In addition to vegetable intake, the average daily energy consumption data among participants indicate an alignment with the general dietary recommendations in Latvia. Interestingly, this intake has seen an increase of approximately 11.2% compared to the data recorded in 2012. During dietary surveys in 2012, women in Latvia reported an average daily caloric intake of about 1728 kcal [24]. The increase in energy intake confirms trends in overweight statistics, suggesting a link between increased caloric consumption and the rise in average body weight observed in recent years [32].

While energy intake is crucial for a healthy diet, macronutrients play a noticeable role. The distribution of macronutrients reveals notable deviations from established dietary guidelines [27], which remarkably affect health conditions [39].

There are differences between the recommended intake of fats and carbohydrates—fat consumption is within the advised range, but carbohydrate intake needs to be higher. Protein intake remains within the recommended range, but fiber intake is below the dietary guidelines [27]. The low fiber intake, as fibers are the primary sources of grains, fruits, berries, and legumes, is established in the study participants’ nutrition. This shortfall in fiber may contribute to the rise in LDL-C levels. Dietary fiber helps lower cholesterol levels by reducing cholesterol absorption in the intestines. A deficiency may contribute to higher levels of LDL-C and an increased risk of CVDs [40].

The participants’ fat intake exceeded the advised range. Such imbalances may increase the risk of CVDs and metabolic disorders, as highlighted in prior research linking high saturated fat intake to adverse lipid profile [28]. This discrepancy raises concerns about an increased risk of CVDs and metabolic disorders, emphasizing the importance of reducing high SF intake to support better lipid profiles.

The consumption of SFA has been associated with a higher risk of CVDs, likely due to the resulting rise in LDL-C [41,42]. However, research indicates that simply lowering SFA intake, without considering the nutrients used as replacements, might have minimal or no impact on disease risk. Replacing saturated fats with PUFA can lead to a decreased occurrence of CVDs [38,43]. Our participants reported a daily intake of PUFA at a level of 7.37%. This percentage falls well within the recommended guidelines the Nordic Council of Ministers set forth, which suggest that PUFA should comprise 5–10% of the total energy consumption [26]. These recommendations are designed to promote optimal health and align with dietary standards to reduce the risk of chronic diseases and improve overall well-being. The participants’ adherence to the recommended PUFA intake reflects positive dietary habits, aligning with evidence suggesting that PUFA consumption reduces CVD risk and promotes optimal health.

Participants’ daily consumption of MUFA adheres to the recommendations set forth by the Nordic Council of Ministers [28]. Although MUFA intake does not significantly affect LDL-C levels, it is linked to a better lipid profile—lower TG levels and higher levels of HDL-C [44,45]. The participants’ adherence to MUFA recommendations supports a favorable lipid profile, mainly through reduced triglyceride levels and increased HDL-C, further reducing CVD risk.

Participants’ carbohydrate intake falls below the recommended minimum [27]. Participants’ carbohydrate intake below the recommended minimum suggests a dietary imbalance that may contribute to suboptimal energy levels and nutritional deficiencies. Increasing carbohydrate intake with whole grains and fiber-rich sources is essential for meeting energy demands and improving dietary quality. Comparing Latvia’s carbohydrate intake with the European average (38–56%), consumption in Latvia is lower than the European average [46]. Latvia’s carbohydrate consumption, being lower than the European average, highlights regional dietary differences and underscores the need to address these gaps to align with broader nutritional standards for better health outcomes.

A low carbohydrate intake has been associated with a higher risk of CVDs, particularly when it leads to the increased consumption of SFA and reduced intake of fiber-rich foods [47]. Research indicates that severely limiting carbohydrates can have a negative effect on heart health by changing the lipid profile, including a rise in LDL-C [48]. Some studies have shown that low-carbohydrate diets effectively promote weight loss and enhance HDL-C and TG levels, but the long-term effects of this nutritional strategy can lead to increased LDL-C and TC. The impact is primarily a result of the dietary composition of low-carbohydrate diets, which often includes increased consumption of SFA [49]. While low-carbohydrate diets may yield short-term benefits like weight loss and improved HDL-C and TG levels, their potential to increase LDL-C and TC levels over the long term indicates a need for cautious implementation and dietary balance.

Low carbohydrate consumption causes participants to have low fiber intake. Dietary fiber is a polymer that exists in two forms: insoluble and soluble. Insoluble fiber helps to increase stool bulk and moves through the digestive system without being digested. Soluble fiber is resistant to digestion and can be fermented by bacteria in the colon into short-chain fatty acids. Research has demonstrated that consuming soluble fiber can enhance the lipid profile [50,51]. The observed low fiber intake among participants is concerning, as dietary fiber is critical in improving lipid profiles, mainly by reducing LDL-C levels and enhancing cardiovascular health.

Protein intake falls within the recommended range, suggesting balanced protein consumption [27]. Latvia’s population protein intake aligns with the European average of (12–20%) and The Nordic Nutrition Recommendations [28,46]. Proteins are essential for human health, but an excessive intake, particularly from animal-origin sources (except fish) of SFA, may negatively affect cardiovascular health. High protein consumption is associated with increased levels of LDL-C [52]. Balanced protein intake within the recommended range is a positive finding; however, caution is needed to ensure that protein sources are not predominantly high in SFA, as this could undermine cardiovascular health benefits.

The findings highlight significant differences between the recommendations and actual dietary fat and carbohydrate intake among participants, which could have long-term implications for the risk of CVDs. Reducing SFA while increasing fiber and plant-origin protein consumption can lower LDL-C by 5–10% [53]. The discrepancies between the recommended and actual dietary fat and carbohydrate intakes suggest a need for targeted nutritional interventions, such as reducing SFA and increasing fiber and plant-based protein intake, which could effectively lower LDL-C and reduce CVD risk.

The average daily intake of vegetables falls short of the guidelines set by the Ministry of Health. Additionally, the recommendations from the Nordic Council of Ministers highlight that individuals should aim to consume between 500 to 800 g of vegetables, fruits, and berries each day; unfortunately, this target is not being achieved [28]. This shortfall suggests a need for increased awareness and efforts to promote healthier eating habits in the population.

A diet low in vegetables is linked to an increased risk of CVDs. Vegetables provide essential nutrients and fiber that help regulate blood pressure and cholesterol levels [54]. In contrast to vegetable consumption, the average daily intake of fruits and berries aligns with the guidelines. The consumption of fruits and vegetables is linked to lower levels of LDL-C, and this connection remains consistent irrespective of factors such as age, smoking status, physical activity, educational background, and vitamin use [55].

The food group “Sugars and related food” demonstrated a strong association with higher levels of remnant cholesterol and non-HDL-C. Research has shown that high levels of remnant cholesterol are connected to a greater risk of heart attacks and strokes [56].

Studies have shown that non-HDL-C is a strong marker for cardiovascular events and a more reliable measure of CVD risk than LDL-C [57].

The FFQ food group “Sugars and related food” contains products with high amounts of added sugars, which, according to the WHO recommendations, must be limited to 10% of total daily energy [58]. There is no mandatory labeling for added sugars in the European Union (EU), including Latvia. As outlined in Regulation (EU) No 1169/2011, which focuses on providing food information to consumers, food labels are required to specify the total amount of sugars in a product. The total sugar [59] is listed under the carbohydrates section of the nutrition label and includes all monosaccharides and disaccharides, irrespective of their origin. Thus, no added sugar intake amount can be calculated for participants.

The food group “Juices” is strongly associated with higher levels of increased LDL-C and non-HDL-C. According to Council Directive 2001/112/E definitions, fruit juice is a natural, unfermented liquid extracted from fruits, maintaining the fruit’s characteristic color, flavor, and taste. Fruit juice from concentrate is made by rehydrating concentrated fruit juice, and the final product must be equivalent to freshly extracted juice in terms of quality and taste [60].

Fruit nectar is obtained by adding water, sugars, and/or honey to the fruit juice, fruit juice from concentrate, and dehydrated/powdered fruit juice to fruit puree or a mixture of those products. Adding sugars and/or honey is permitted up to 20% of the total weight of the finished product [60].

Unlike nectars, juice drinks contain a smaller amount of juice or concentrated juice, and laws and regulations do not set the amounts—vegetable juice contains a minimal amount of sugar but a high salt level.

Consuming fruit beverages can lead to the overconsumption of natural and added sugars. High added-sugar consumption is linked to a greater risk of dyslipidemia, resulting in increased TG levels, reduced HDL-C, and heightened inflammation, contributing to atherosclerosis [61].

The intake of sweetened drinks may decrease the consumption of solid foods, but not to an extent that prevents an overconsumption of calories, which leads to a gradual rise in obesity. When equal amounts of sucrose or maltose are added to solid foods used in the diet, they do not have the same effect. Consequently, studies suggest that sweetened beverages increase caloric intake [62].

The FFQ food group “Sweets” demonstrated a strong association with elevated levels of remnant cholesterol and non-HDL-C. The food group included the following products: biscuits, cakes, chocolate, halva, sherbert, cereal bars, lollipops, caramels, marshmallows, marmalade, pastilles, ice cream/sorbet, sweet popcorn, and confectionery. This food group can be described as high-processed products with high amounts of added sugars.

High-fructose corn syrup (HFCS) has become very popular in producing different products and is one of the primary sources of calories in the diet. HFCS has mostly replaced sucrose in the food industry. The combination of fructose and glucose in HFCS increases LDL-C. This suggests that HFCS may be as harmful as pure fructose, highlighting the need for strategies to reduce free-sugar intake [63]. There is no straightforward way to gauge the overall consumption of HFCS because studies and dietary surveys need to differentiate between HFCS and other added sugars, especially on food labels. The formulations of HFCS differ in their ratios of fructose to glucose and their total sugar content [62]. Although HFCS is present in most processed food, manufacturers do not provide specific composition details, nor are they available in public food composition databases [37,64].

Higher meat and poultry consumption showed a negative association with TC levels. The data from other research support the study results. A meta-analysis proved that red meat consumption affected TG but had a minimal impact on TC, LDL-C, and HDL-C [65,66]. No correlations have been found between the intake of poultry and CVD risk, and poultry is suggested to be a healthier alternative to processed and red meat [67].

However, it is also essential to consider the overall dietary context. The observed correlations may be influenced by several dietary factors, particularly the types of meat consumed and their preparation methods. This study’s data indicated that consuming lean red meat from a specific breed did not negatively impact CVD risk compared to lean white meat. By choosing lean cuts, individuals can reduce their intake of SFA, which may improve their lipid profile [68]. The significance of cooking techniques (raw preparation, boiling, roasting, and pan-frying) and their possible advantages for cardiovascular health were observed in several studies [69,70]. The regular consumption of fried foods has been found to have a significant association with elevated cholesterol levels in the blood [71]. Consuming lean meats and skinless poultry while avoiding processed meats like sausages could contribute to the observed favorable lipid profiles.

The study presents several limitations that should be considered. Firstly, the cross-sectional design limits the ability to determine causal relationships between dietary patterns and risk factors for CVDs. Future research using longitudinal or intervention designs could provide more substantial evidence. Secondly, the sample size of 61 participants restricts the generalization of the findings. Therefore, viewing these results as part of an exploratory investigation is essential.

Additionally, using an FFQ to assess dietary intake may introduce recall bias, leading to inaccuracies in the reported data. Moreover, the study sample consisted of participants recruited from private clinics, including educated and motivated individuals, potentially introducing selection bias. A comparative analysis with data from larger population studies is necessary to determine whether similar dietary patterns, BMI, and other relevant factors are observed in the broader population of women in the same age group. Furthermore, the study did not extensively evaluate the influence of physical activity levels, a critical factor known to affect cholesterol and overall cardiovascular health. Including such variables in future research could provide a more comprehensive understanding of the interactions between diet, physical activity, and lipid profiles. Additional considerations on these aspects could further strengthen the analysis.

## 5. Conclusions

This study highlights the impact of high-sugar diets on the development of an unfavorable lipid profile, which serves as a risk marker for CVDs. Conversely, the observed negative association between the consumption of meat and poultry suggests potential protective effects against these risks. It is essential to consider the broader dietary context, as the effects can vary depending on the type of meat consumed and the methods used for its preparation.

These findings emphasize the complex relationship between specific dietary patterns and factors that influence the risk of CVDs. However, the results should be interpreted with caution due to certain limitations. Future research involving larger populations should consider physical activity levels and other dietary factors to provide a more comprehensive understanding of the interactions between diet, lifestyle factors, and lipid profiles.

## Figures and Tables

**Figure 1 nutrients-17-00243-f001:**
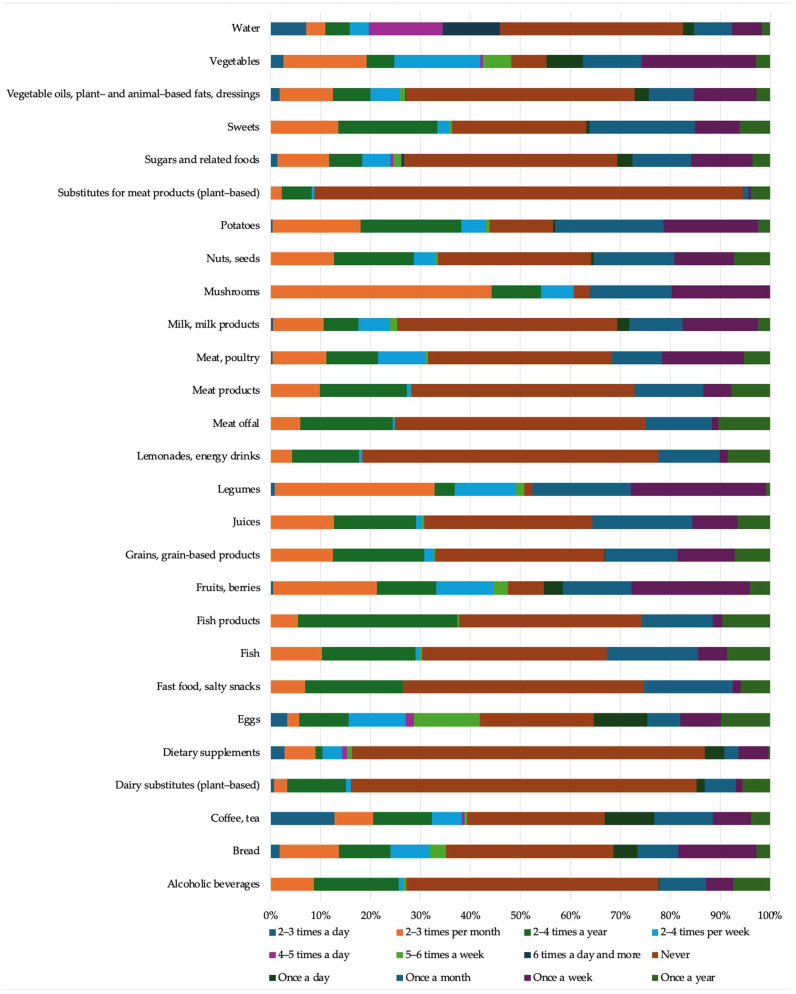
Average food and beverage intake. Data from the food frequency questionnaire (*n* = 61).

**Figure 2 nutrients-17-00243-f002:**
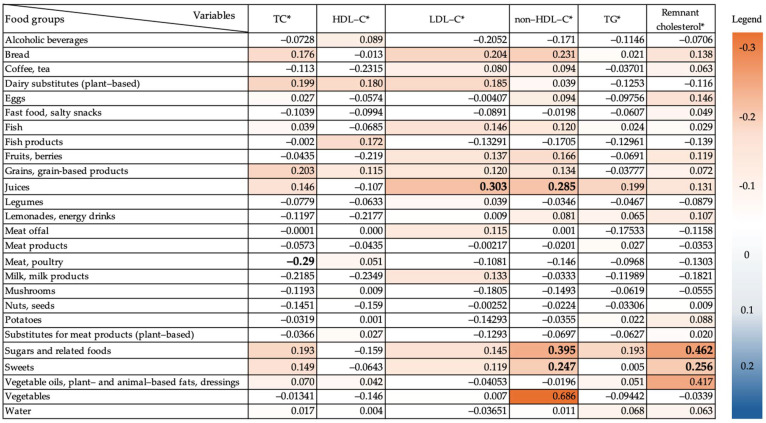
Heat map of standardized regression coefficients (β) obtained from linear regression analyses between median food group intakes and lipid profile parameters. * Adjusted by age, BMI, and waist-to-hip ratio. Standardized regression coefficients (β) with *p*-value ≤ 0.05 are shown in bold.

**Table 1 nutrients-17-00243-t001:** Characteristics of the participants (*n* = 61).

Characteristics	
Age (years)	49 ± 3
Body mass index (kg m^−2^)	25.05 ± 4.92
Waist circumference (cm)	84.34 ± 11.26
Waist-to-hip ratio	0.81 ± 0.07
Use of alcohol (yes)	36 (59.02%)
Smoking (yes)	3 (4.92%)
Daily light physical activity *	51 (83.61%)
Number of main meals per day:	
<2	12 (19.67%)
2–3	45 (73.77%)
4–5	4 (6.56%)
Number of snacks per day:	
no	2 (3.28%)
1	20 (32.79%)
2	23 (37.70%)
3	12 (19.67%)
≥4	4 (6.56%)

* walking, moving around without a car, climbing stairs, house cleaning.

**Table 2 nutrients-17-00243-t002:** Biochemical profile of the participants (*n* = 61).

Parameters	Mean ± SD	Reference Range
TC (mmol L^−1^)	5.93 ± 0.54	0.0–5.0
HDL-C (mmol L^−1^)	1.85 ± 0.46	>1.2
LDL-C (mmol L^−1^)	3.61 ± 0.30	1.7–3.0
TG (mmol L^−1^)	1.15 ± 0.64	0.1–1.7
ALAT (U L^−1^)	21.00 ± 10.21	10–35
Glucose (mmol L^−1^)	4.67 ± 0.67	4.1–5.9

**Table 3 nutrients-17-00243-t003:** Average daily intake of food products and beverages (*n* = 61).

Food Products or Beverages	Average Daily Intake ± SD, g/Day
Grains, grain-based products	74.55 ± 10.01
Bread	68.69 ± 18.57
Potatoes	44.86 ± 24.18
Meat, poultry	89.45 ± 21.86
Meat offal	5.46 ± 1.42
Meat products	14.99 ± 2.73
Eggs	51.41 ± 45.18
Fish	38.67 ± 6.85
Fish products	6.23 ± 3.29
Milk, milk products	305.71 ± 32.33
Vegetable oils, plant- and animal-based fats, dressings	27.43 ± 4.69
Substitutes for meat products (plant-based)	1.42 ± 2.71
Dairy substitutes (plant-based)	52.23 ± 59.91
Vegetables	203.62 ± 25.09
Fruits, berries	232.85 ± 33.85
Mushrooms	4.93 ± 4.89
Legumes	17.80 ± 15.38
Nuts, seeds	20.54 ± 2.64
Sweets	46.48 ± 6.18
Sugars and related foods	6.95 ± 3.25
Fast food, salty snacks	29.00 ± 3.85
Lemonades, energy drinks	17.46 ± 10.52
Juices	39.88 ± 19.55
Alcoholic beverages	45.04 ± 13.13
Coffee, tea	741.57 ± 221.59
Water	1187.20 ± 565.44

**Table 4 nutrients-17-00243-t004:** Average daily energy and macronutrient intakes among participants (*n* = 61).

Nutrients	Average Daily Intake	Guidelines for Recommended Daily Intake
Energy, kcal	1921.67 ± 20.64	1840–2360 kcal [27]
Proteins, g	94.06 ± 1.37	
Proteins E%	19.58%	10–20% [27]
Carbohydrates, g	169.81 ± 2.32	
Carbohydrates E%	35.35%	45–60% [27]
Total sugars, g	4.67 ± 0.18	
Fiber, g	22.39 ± 0.38	25–35 g [27]
Soluble fiber, g	6.82 ± 0.13	
Insoluble fiber, g	14.53 ± 0.25	
Total fats, g	90.55 ± 1.37	
Total fats E%	42.41%	20–30% [27]
Saturated fatty acids, g	28.86 ± 0.44	
Saturated fatty acids E%	13.52%	< 10% [27]
Monounsaturated fatty acids, g	34.42 ± 0.68	
Monounsaturated fatty acids E%	16.12%	10–20% [28]
Polyunsaturated fatty acids, g	15.73 ± 0.30	
Polyunsaturated fatty acids E%	7.37%	5–10% [28]

## Data Availability

The data supporting this study’s findings are available upon request from the corresponding author [S.A.]. The data are not publicly available due to containing information that could compromise the privacy of research participants.

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
