# Peer review of "Plasma Lipid Profile Among Perimenopausal Latvian Women in Relation to Dietary Habits"

_nutrients, 2025, doi:10.3390/nu17020243_

Round 1
Reviewer 1 Report
Comments and Suggestions for Authors
The current study evaluated the association between dietary habits and plasma lipid profile among perimenopausal women in Latvia.
Some comments:
1. page 2, line 46 – what about 2024? Please add.
2. Please add at suppl material the questionnaire model. At point 2.4. add please who made the questionnaire .
3. Please clarify how many women where involved in the study. At line 13, abstract you wrote “The randomised clinical trial involved perimenopausal women (n = 61)”, at page 3, line 117-18 that “(n = 108) remained eligible” and at line 130 that “two groups with 30 participants in every group were defined” . Finally at line 162 you wrote: “In total, 61 (sixty-one) participants completed the study”.
If 108 were eligible what happened to the others 47? In my opinion the number of the women included into the study is too small.4. page 3 – It’s not clear how the randomization of participants was done. Please reformulate.
5. page 7, line 225 – Please clarify what do you mean by “The participants' average daily energy intake aligns with the recommendations”
6. Please explain why did you choose to determine blood glucose and ALAT and how high-sugar diets or high-proteic diet can influence the lipid profile.
7. TC, LDL-C, HDL-C, TG, 120 ALAT, and glucose levels were determined from plasma? Please clarify.
8. Discussion: please discuss the results of your study, not just present aspects taken from the literature
Reviewer 2 Report
Comments and Suggestions for Authors
The manuscript presents interesting results from a study of plasma lipid profile among perimenopausal women in relation to their dietary habits. I appreciate the effort to reach as many potential study participants as possible, even though the data were finally obtained only from 61 women. This shows both the complexity of such studies and the valuable information they provide.
The text is properly structured, the Introduction provides insight into the issue, the methodology is described in detail, the results are clearly presented and properly discussed, the Conclusion summarizes the obtained outputs. I therefore have only some minor comments about the manuscript:
L 46: Do you have information what is the proportion of women among the death caused by CVDs?
L 50: Explain the abbreviation TH.
L 121: Please specify the accredited laboratory (accreditation number, accreditation body, place of the laboratory).
L 124: Please specify Central Laboratoria (place).
L 169: Please specify the used software (producer, town, country).
Table 3: It would be good to explain in the text what some categories exactly cover:
Meat, poultry – why is poultry not automatically considered as meat? Why is fish not considered as meat?
Why are vegetable oils in one category with animal fats? They could have significantly different proportion of SFA/MUFA/PUFA and effect to plasma lipid profile.
What is the difference between “sweets” and “sugars related foods”?
What do you consider as “fast food”? Do you include fast food meat product to this category as well as to the category “meat products”?
“Salty snacks” could also belong to the categories such as potato, grain-based products, or nuts.
L 199-218: Consider how much you need to repeat in the text the values that are already given in the tables?
L 254: TH levels are not given in Figure 2.
L 255: Please add the information about the correlation between vegetables and non-HDL-C.
L 266-275: Consider how much you need to repeat in the text the values that are already given in the tables?
I understand that for women in Latvia the recommendations of the Ministry of Health of the Republic of Latvia or Nordic Council of Ministers are the most relevant. But have you tried to compare your results also with other recommendations such as DACH, EFSA, WHO recommendations?
L 333: Fish as animal origin source do not contain high levels of SFA, but significant levels of PUFA.
L 386-390: The specification of each group should be given in Materials and Methods.
Round 2
Reviewer 2 Report
Comments and Suggestions for Authors
I would like to thank the authors for considering my comments and answering my questions, which, in my opinion, improved the overall quality of the manuscript. After careful reading of the answers to my questions I can recommend accepting the manuscript by Nutrients. No comments on the revised manuscript.